# The Dipeptidyl Peptidase-4 Inhibitor Linagliptin Ameliorates Endothelial Inflammation and Microvascular Thrombosis in a Sepsis Mouse Model

**DOI:** 10.3390/ijms23063065

**Published:** 2022-03-12

**Authors:** Shen-Chih Wang, Xiang-Yu Wang, Chung-Te Liu, Ruey-Hsing Chou, Zhen Bouman Chen, Po-Hsun Huang, Shing-Jong Lin

**Affiliations:** 1Cardiovascular Research Center, National Yang Ming Chiao Tung University, Taipei 11221, Taiwan; akkwang@gmail.com (S.-C.W.); thanatosrs@gmail.com (R.-H.C.); sjlin@vghtpe.gov.tw (S.-J.L.); 2Department of Anesthesiology, Taipei Veterans General Hospital, Taipei 11217, Taiwan; 3Institute of Clinical Medicine, National Yang Ming Chiao Tung University, Taipei 11221, Taiwan; yuh0817ovo@gmail.com; 4Division of Nephrology, Department of Internal Medicine, Wan Fang Hospital, Taipei 116, Taiwan; 96320@w.tmu.edu.tw; 5Department of Internal Medicine, School of Medicine, College of Medicine, Taipei Medical University, Taipei 11031, Taiwan; 6Department of Critical Care Medicine, Taipei Veterans General Hospital, Taipei 11217, Taiwan; 7Division of Cardiology, Department of Internal Medicine, Taipei Veterans General Hospital, Taipei 11217, Taiwan; 8Department of Diabetes Complications and Metabolism, City of Hope, Duarte, CA 91010, USA; 9Irell and Manella Graduate School of Biological Sciences, City of Hope, Duarte, CA 91010, USA; 10Department of Medical Research, Taipei Veterans General Hospital, Taipei 11217, Taiwan; 11Division of Cardiology, Heart Center, Cheng-Hsin General Hospital, Taipei 11220, Taiwan; 12Taipei Heart Institute, Taipei Medical University, Taipei 11031, Taiwan

**Keywords:** dipeptidyl peptidase-4 inhibitor, inflammation, linagliptin, sepsis, thrombosis

## Abstract

The pathophysiology of sepsis involves inflammation and hypercoagulability, which lead to microvascular thrombosis and compromised organ perfusion. Dipeptidyl peptidase (DPP)-4 inhibitors, e.g., linagliptin, are commonly used anti-diabetic drugs known to exert anti-inflammatory effects. However, whether these drugs confer an anti-thrombotic effect that preserves organ perfusion in sepsis remains to be investigated. In the present study, human umbilical vein endothelial cells (HUVECs) were treated with linagliptin to examine its anti-inflammatory and anti-thrombotic effects under tumor necrosis factor (TNF)-α treatment. To validate findings from in vitro experiments and provide in vivo evidence for the identified mechanism, a mouse model of lipopolysaccharide (LPS)-induced systemic inflammatory response syndrome was used, and pulmonary microcirculatory thrombosis was measured. In TNF-α-treated HUVECs and LPS-injected mice, linagliptin suppressed expressions of interleukin-1β (IL-1β) and intercellular adhesion molecule 1 (ICAM-1) via a nuclear factor-κB (NF-κB)–dependent pathway. Linagliptin attenuated tissue factor expression via the Akt/endothelial nitric oxide synthase pathway. In LPS-injected mice, linagliptin pretreatment significantly reduced thrombosis in the pulmonary microcirculation. These anti-inflammatory and anti-thrombotic effects were independent of blood glucose level. Together the present results suggest that linagliptin exerts protective effects against endothelial inflammation and microvascular thrombosis in a mouse model of sepsis.

## 1. Introduction

Sepsis, defined as infection with consequent systemic inflammatory response syndrome, is a major cause of critical illness and mortality. As sepsis progresses, dysregulation between pro- and anti-coagulation pathways leads to hypercoagulability and compromised organ perfusion [1,2]. The pathophysiology of severe sepsis entails multi-system organ dysfunction, including shock; injury to lung, kidney, and liver; coagulopathy [3,4]. Currently, there are only limited treatments for sepsis, such as source control, volume repletion, vasoconstrictor use, and ventilator support [5,6]. Pharmaceutical treatments to intervene in the dysregulation of sepsis are lacking. Thus, investigation of infection-induced activation of coagulation may lead to new therapeutic strategies for the treatment of sepsis.

The vascular endothelium is a cellular monolayer lining the luminal surface of blood vessels. During sepsis, endothelial cells (ECs) respond to various pro-inflammatory stimuli and trigger downstream sepsis-related processes. For example, in response to lipopolysaccharide (LPS) or pro-inflammatory cytokines, ECs express pro-inflammatory molecules that contribute to endothelial dysfunction and a pro-coagulation status [7,8]. Tumor necrosis factor (TNF)-α has been shown to activate the nuclear factor (NF)-κB pathway in ECs to induce the expression of intercellular adhesion molecule (ICAM)-1 and procoagulant tissue factor (TF) [9,10]. The activation of NF-κB also suppresses phosphorylation of endothelial nitric oxide synthase (eNOS) and the production of nitric oxide (NO) [11]. NO production in the endothelium inhibits platelet aggregation and leukocyte adhesion, and contributes to the maintenance of EC function [12,13]. Additionally, the endothelial production of NO has inhibitory effects on venous thrombosis and inflammation [14,15]. Collectively, ECs play a central role in the pathophysiology of sepsis and may be a plausible therapeutic target.

Dipeptidyl peptidase (DPP)-4 inhibitors are widely used for the treatment of type 2 diabetes mellitus. Emerging evidence shows that these drugs exert an anti-inflammatory effect, and thus may protect against inflammatory conditions. Hwang et al. [16] reported that a DPP-4 inhibitor attenuated the LPS-induced expression of vascular adhesion molecules and pro-inflammatory cytokines via Akt- and AMP-activated protein kinase-dependent signaling pathways. In animal models of endotoxemia, DPP-4 inhibitors reduced oxidative stress, mitigated platelet activation, and suppressed renal fibrosis [17,18]. These findings suggest that DPP-4 inhibitors may disrupt the pathophysiology of sepsis and offer promise as adjuvant treatments. However, no clinical investigations have yet examined such use, and the effects of DPP-4 inhibitors on TF expression, thrombosis, and organ perfusion remain uncertain. In the present study we examined the anti-inflammatory and anti-thrombotic effects of linagliptin, a DPP-4 inhibitor, on in vitro and in vivo models of sepsis. We also explored the signaling pathways underlying these effects.

## 2. Results

### 2.1. Linagliptin Attenuated TNF-α–Induced Inflammatory Signaling In Vitro

TNF-α treatment significantly increased mRNA levels of the NF-κB p65 subunit in HUVECs, simulating the upregulated inflammation that occurs in sepsis. Linagliptin treatment (1 and 10 μM) significantly attenuated this effect (*p* = 0.0217 and 0.0043, respectively, Figure 1A). Similarly, the TNF-α-induced nuclear translocation of NF-κB was significantly attenuated by linagliptin (1 and 10 μM) (*p* = 0.0421 and 0.0331 respectively, Figure 1B). Furthermore, TNF-α-induced expression of IL-1β and ICAM-1 was significantly inhibited by linagliptin (1 and 10 μM) (for IL-1β, *p* = 0.0107 and 0.0478, respectively; for ICAM-1, *p* = 0.0046 and 0.02, respectively, Figure 1C,D).

### 2.2. Linagliptin Suppressed the TNF-α–Induced Expression of TF in HUVECs via the Akt/eNOS Pathway

We also examined the effect of linagliptin on TF and eNOS. TNF-α treatment significantly increased TF expression, an effect that was ameliorated by linagliptin (1 and 10 μM) (*p* = 0.0103 and 0.0070, respectively, Figure 2A). TNF-α-induced suppression of eNOS phosphorylation was reversed by linagliptin treatment, even beyond the level observed without TNF-α treatment (*p* = 0.0189 and 0.0047, respectively, Figure 2B). TNF-α treatment did not significantly change the phosphorylation of Akt, but linagliptin enhanced it (*p* = 0.0085 and 0.0050, respectively, Figure 2C). To confirm the role of eNOS in TNF-α–induced TF expression, we examined whether the eNOS inhibitor L-NAME could block the protective effect of linagliptin. Indeed, the suppressive effect of linagliptin (1 and 10 μM) on TNF-α–induced TF expression in HUVECs was completely abolished by L-NAME treatment (*p* = 0.0044 and 0.251, respectively) (Figure 2D).

### 2.3. Linagliptin Ameliorated LPS-Induced Expression of Inflammatory Cytokines and TF in Vascular Tissue In Vivo

mRNA expression levels of the NF-κB p65 subunit, IL-1β, and ICAM-1 were significantly increased in the aortas of mice with LPS-induced sepsis. Pretreatment with linagliptin significantly attenuated the LPS-induced inflammation cascade in aortic tissue (Figure 3A–C). For untreated control versus LPS-injected mice, the *p*-value of NF-κB p65 subunit was 0.0004; IL-1β was <0.0001, and ICAM-1 was <0.0001. For LPS-injected versus LPS-injected plus linagliptin, the *p*-value of NF-κB p65 subunit was 0.0473, IL-1β was 0.0014, and ICAM-1 was 0.0488. TF expression was increased in the endothelial layer of aortic tissue from mice treated with LPS as well as those treated with LPS plus linagliptin and L-NAME, as evidenced by immunofluorescence staining (Figure 3D).

### 2.4. Linagliptin Treatment Reduced Pulmonary Microvascular Thrombosis and Improved Pulmonary Perfusion in a Mouse Model of LPS-Induced Sepsis

Next, we examined the effect of linagliptin in vivo using a mouse model of LPS-induced sepsis. LPS treatment had no effect on eNOS expression. However, linagliptin treatment significantly increased eNOS expression in mice with LPS-induced sepsis; L-NAME reversed this effect (Figure 4A). For pulmonary microvascular thrombosis detection, fluorescence was detected in mice lung (excitation: λ640 nm; emission: λ680 nm). Fluorescent beads would trap in the embolized vessels result in increased fluorescence signal. Little fluorescence was detected in the lungs of untreated controls; much more fluorescence was observed in the LPS treatment group (*p* = 0.0056 compared to untreated lungs), suggesting sepsis-induced pulmonary thrombosis. Linagliptin treatment significantly reduced the levels of pulmonary fluorescence (*p* = 0.0184 compared to LPS-treated lungs), an effect that was blocked by L-NAME administration (Figure 4B). Blood glucose levels did not differ significantly across any of the treatment groups (Figure 4C). Body weight changes are shown in Appendix A.

## 3. Discussion

The present study showed that linagliptin suppressed TNF-α–induced pro-inflammatory cytokines, adhesion molecules, and pro-thrombotic factors in HUVECs. Our proposed mechanism is shown in Figure 5. In vivo, linagliptin suppressed LPS-induced inflammatory cytokines and adhesion molecules. It also reduced LPS-induced thrombosis and improved pulmonary microcirculation. These findings suggest that linagliptin has protective effects against sepsis and may be suitable as an adjuvant treatment.

Linagliptin and other DPP-4 inhibitors suppress DPP-4, a peptidase which cleaves hypoglycemia incretins secreted by intestinal L cells such as glucagon-like peptide (GLP)-1, thereby lowering the post-prandial blood glucose level [19,20]. In addition to their hypoglycemic effects, DPP-4 inhibitors have also been shown to exert anti-inflammatory and anti-atherosclerotic effects against cardiovascular diseases [21,22] and an anti-inflammatory effect in experimental models of sepsis [17,23]. In accordance with these findings, the present study showed that linagliptin suppressed TNF-α–induced inflammation in vitro and LPS-induced inflammation in vivo, suggesting its potential to be an adjuvant treatment for sepsis and other inflammation-related diseases.

As oral antidiabetic drugs, DPP-4 inhibitors may exert their anti-inflammatory effects via a hypoglycemic effect; hyperglycemia has been associated with proinflammatory responses [24,25]. However, since our in vitro results were obtained from HUVECs, which do not secrete GLP-1 or other incretins, the anti-inflammatory and anti-thrombotic effects we observed were completely independent of any hypoglycemic effect. In addition, the blood glucose level did not differ among mouse groups in our in vivo experiments. Taken together, this evidence suggests that the anti-inflammatory and anti-thrombotic effects of linagliptin are independent of glucose control [23].

TNF-α stimulation is known to activate NF-κB, which translocates into the nucleus to trigger transcription associated with inflammation and thrombosis [26,27]. NF-κB is a transcription factor that triggers the expression of pro-inflammatory cytokines such as IL-1β and leukocyte adhesion molecule ICAM-1 to augment the inflammation of involved tissues, in a process referred to as inflammatory cascade. In the present study, linagliptin down-regulated the expression of IL-1β and ICAM-1 by suppressing the nuclear translocation of NF-κB in a HUVEC model of TNF-α–induced inflammation. This result suggests that linagliptin may offer benefits against other NF-κB–related inflammatory conditions.

Interaction between TF and coagulation factor VIIa activates the coagulation cascade [28]. TF expression is suppressed by eNOS, which is activated via Akt-dependent signaling pathways [29,30]. In our in vitro experiments, linagliptin increased the phosphorylation of Akt at Ser-473, which is critical for eNOS activation and NO production. As Akt was activated under linagliptin treatment, eNOS phosphorylation at Ser-1177 increased and TF expression was subsequently suppressed. In our in vivo experiments, linagliptin treatment blocked the formation of LPS-induced microvascular thromboses. These findings suggest that linagliptin exerts an anti-thrombotic effect via the Akt/eNOS pathway and might help to preserve organ perfusion in patients with sepsis.

Thrombosis is a key factor leading to organ hypoperfusion and dysfunction in sepsis. To quantify microvascular thrombus formation resulting from sepsis-induced disseminated intravascular coagulation (DIC), the intravenous injection of fluorescent beads was used. This technique was established in previous studies in healthy hairless mice, in which consecutive fluorescence detection showed that the beads passed through the pulmonary circulation and accumulated in the spleen and liver [17,31]. We obtained similar findings for the lungs of healthy control mice, and observed markedly increased pulmonary fluorescence in mice with LPS-induced sepsis, indicating the successful induction of DIC in our mouse model. The present results, in a mouse model of LPS-induced sepsis, suggest that linagliptin exerted an anti-thrombotic effect via the eNOS-dependent signaling pathway.

One limitation of our study is that we did not establish a model of sepsis induced by real pathogens, which limits the application of our findings to patients in the clinical context. Another limitation is that we did not test linagliptin at a lower concentration; as low as 50 nM has shown protective effects in vitro [32,33]. Additional work is needed to investigate the potential toxicity or benefits of different linagliptin concentrations. Moreover, we used L-NAME to interrogate the role of eNOS phosphorylation in linagliptin’s protective effects. To complete the axis of p-Akt/p-eNOS/NO signaling that we proposed, further investigation involving inhibition of Akt phosphorylation is warranted. Finally, eNOS played a complicated role in sepsis. Reduced eNOS activity in septic mice contributed to poorer outcome [34]. In an observational study, sepsis perturbated endothelial function and NO production [35]. Microvascular dysfunction due to reduced eNOS activity and NO synthesis also appears to play an important role in outcome [36]. Our study was not designed to elucidate such a complicated network.

In conclusion, this study showed that the anti-inflammatory and anti-thrombotic effects of linagliptin are independent of glucose control in vitro and in vivo. Linagliptin indirectly suppresses thrombosis via the Akt/eNOS pathway and exerts anti-inflammatory effects via NF-κB signaling. Our findings support consideration of linagliptin as a potential adjuvant treatment to protect against sepsis-induced DIC. Such a hypothesis warrants future testing in a clinical setting.

## 4. Materials and Methods

### 4.1. Cell Experiments

Human umbilical vein endothelial cells (HUVECs) were obtained from the American Type Culture Collection (Manassas, VA, USA) and cultured in Endothelial Cell Growth Basal Medium^TM^-2 (Lonza, Basel, Switzerland) with 10% fetal bovine serum, 100 units/mL penicillin, and 100 mg/mL streptomycin under standard conditions (37 °C, 95% humidified air, and 5% CO_2_). Subcultures were performed with trypsin-EDTA. The media were refreshed every 2 days. Experiments were conducted first to examine the effect of linagliptin after the confirmation of cell viability. In brief, different doses of linagliptin (1, 10, and 100 µM, dissolved in PBS; SantaCruz Biotechnologies, Santa Cruz, CA, USA) were administered to HUVECs for 48 h. The cells were then treated with 3-(4,5-dimethylthiazol-2-yl)-2,5, diphenyltetrazolium bromide (0.5 mg/mL; Sigma-Aldrich, St. Louis, MO, USA) for 4 h and lysed with dimethyl sulfoxide, and their absorbance was measured at 550/650 nm by the Sunrise ELISA plate reader (TECAN, Männedorf, Switzerland). Compared with the control group, 1 and 10 µM linagliptin had no effect on cell viability, but 100 µM linagliptin reduced HUVEC viability by 30%. Thus, we treated HUVECs with 1 and 10 µM linagliptin for 48 h in subsequent in vitro studies.

To simulate a sepsis environment, HUVECs were treated with 0.5 ng TNF-α for 30 min, with or without linagliptin pretreatment. For experiments involving NO inhibition, the NO inhibitor NG-nitro-L-arginine-methylester (L-NAME; Sigma-Aldrich, St. Louis, MO, USA) 1mM was administered 1 h before linagliptin administration.

### 4.2. Western Blot Analysis

HUVECs were washed with cold phosphate-buffered saline and lysed with RIPA buffer (1 mM phenylmethylsulfonyl fluoride, 2 µg/mL leupeptin, aprotinin, and pepstatin). Cell lysates were centrifuged, and the supernatants harvested as protein samples. For experiments involving nuclear proteins, cytosolic and nuclear proteins were separated using a nuclear/cytosol fractionation kit (BioVision, Milpitas, CA, USA) and their concentrations were measured using a Bradford protein assay (Bio-Rad, Hercules, CA, USA). For each sample, 40 μg protein was loaded onto 10% sodium dodecyl sulfate-polyacrylamide gels and then transferred to nitrocellulose membranes using iBlot™ 2 transfer stacks (Invitrogen, Carlsbad, CA, USA) for 6 min at 19 V. The membranes were blocked with 3% bovine serum albumin (Sigma-Aldrich, St. Louis, MO, USA) for 1 h at room temperature, then incubated with primary antibodies overnight at 4 °C. The primary antibodies used in the present study were p-eNOS (ser1177), eNOS, p-Akt, Akt, β-actin, and lamin B (details and sources are provided in Appendix A). The nuclear translocation of NF-κB was examined further using the nuclear/cytosolic NF-κB protein ratio, with the nuclear protein lamin B serving as a nuclear marker.

After appropriate washing, the membranes were incubated with secondary antibodies for 1 h at room temperature. Blotting signals were detected using chemiluminescence detection reagents (PerkinElmer, Waltham, MA, USA).

### 4.3. RNA Extraction and Quantitative Real-Time Polymerase Chain Reaction

Total RNA was extracted using a NucleoZOL kit (Macherey-Nagel, Duren, Germany). To confirm RNA quality, concentrations were determined by measuring the 260/230 and 260/280 values with a NanoDrop device (Thermo Fisher Scientific, San Diego, CA, USA).

Prior to quantitative polymerase chain reaction (qPCR), the RNA samples were transferred to complementary (c)DNA. In brief, total RNA was transferred to single-stranded cDNA using the PrimeScript™ RT reagent kit (Takara, Otsu, Shiga, Japan). One reaction volume included 2 μL 5X PrimeScript buffer, 0.5 μL PrimeScript RT enzyme mix, 0.5 μL Oligo dT primer (50 μM), and 0.5 μL random 6 mers (100 μM). These ingredients were mixed with 500 ng total RNA in RNase-free water, then transferred to the PCR tube. Reverse transcription was performed with cycles of 37 °C for 15 min, 85 °C for 5 s, and holding at 4 °C. The cDNA was stored at –20 °C until further analysis.

For real-time PCR, Fast SYBR^TM^ Green Master Mix (Thermo Fisher Scientific, San Diego, CA, USA) was used. One reaction volume included 5 μL Fast SYBR Green Master Mix, 0.5 μL (10 μM) of forward and reverse primers, and 2.5 μL RNase-Free water. These ingredients were mixed with 2 μL 50-ng cDNA. The reaction was performed using an ABI StepOnePlus real-time PCR system (Applied Biosystems Inc., Foster City, CA, USA) at 95 °C for 20 s, followed by 40 cycles of 3 s at 95 °C, 10 s at 62 °C, and 20 s at 60 °C. The average threshold cycle for each gene was normalized to GAPDH for analysis. The primer sequences are provided in Appendix A.

### 4.4. Mouse Model of LPS-Induced Microvascular Thrombosis

Male C57BL/6J mice aged 8 weeks were purchased from the National Laboratory Animal Center, Taiwan, and allowed to acclimate for 2 weeks. Mice were housed in the Institutional Animal Care Committee facility of Taipei Veterans General Hospital with a 12-h day/night cycle and unrestricted access to water and food. Protocols were approved by the Institutional Animal Care and Use Committee of the Taipei Veterans General Hospital. At the time of the experiment, mice were 10–12 weeks old and weighed 25–30 g. The animals were divided into four groups: (1) negative controls; (2) LPS-injected; (3) linagliptin-treated and LPS-injected; (4) L-NAME pre-treated, linagliptin-treated, and LPS-injected. L-NAME treatment began two weeks before LPS injection. Linagliptin treatment (5 mg/kg/day) was administered by oral gavage [37] for 2 days prior to LPS injection. LPS (10 mg/kg) purified from *Salmonella typhosa* was administered by single intraperitoneal injection 24 h before the end of the experiment. Lung tissue and blood were collected for analysis of pulmonary microvascular thrombosis and blood sugar, respectively. Aortic tissues were isolated to determine the expression of mRNA using qPCR.

### 4.5. Fluorescence Imaging of Aortic Endothelial Tissue Factor Expression

Cryosections (6 μm) of aortic tissue were fixed with paraformaldehyde, permeabilized, blocked, and incubated with rat anti–mouse TF antibody (SC-18916, SantaCruz Biotechnologies, Santa Cruz, CA, USA) overnight at 4 °C. Sections were then washed and incubatedwith goat-anti-rat Alexa 568–conjugated secondary antibody (Thermo Fisher Scientific, Rochester, NY, USA) in room temperature for 1 h. After washing 3 times, sections were then incubated with mouse anti–mouse CD31 antibody (SC-46694, SantaCruz Biotechnologies, Santa Cruz, CA, USA) overnight at 4 °C followed by incubation with rabbit-anti-mouse Alexa 488–conjugated secondary antibody (Thermofisher Scientific, Rochester, NY, USA) in room temperature for 1 h. Sections were mounted and viewed under confocal microscope.

### 4.6. Fluorescence Imaging of Pulmonary Microvascular Thrombosis

For the visualization of microvascular occlusion, mice were injected with fluorescent beads (2.5 μM, Invitrogen^TM^) diluted in sterile saline (2.4 × 10^4^ beads/μL, 100 μL/animal) via the tail vein 3 h after LPS injection [17]. At 72 h after linagliptin treatment, the mice were sacrificed and blood, aorta, and lungs collected for analysis. Fluorescence in embolized vessels, enhanced by the fluorescent beads, was quantified using the PhotonIMAGER™ OPTIMA device (Biospace Lab, Paris, France).

### 4.7. Measurement of Interleukin-1β and ICAM-1

Relative fold changes in mRNA expression levels of interleukin (IL)-1β and ICAM-1 were used as a marker of inflammatory signaling in the TNF-α-treated HUVEC and in aortic tissue from LPS-injected mice.

### 4.8. Statistical Analysis

The data are expressed as mean ± standard error. Comparisons among groups were performed using one-way analysis of variance followed by Scheffe’s multiple-comparison post hoc test. All statistical analyses were conducted using SPSS software version 14 (SPSS, Chicago, IL, USA). *P* values < 0.05 were considered significant.

## Figures and Tables

**Figure 1 ijms-23-03065-f001:**
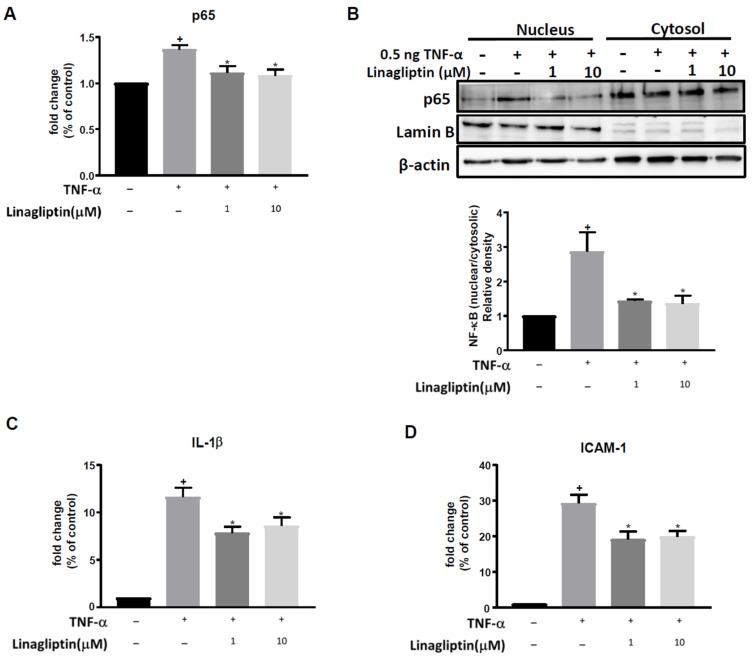
Linagliptin exerts anti-inflammatory effects by suppressing the TNF-α-induced nuclear translocation of NFκB. HUVECs were pretreated with different doses (1 and 10 µM) of linagliptin for 48 h and then incubated with 0.5 ng TNF-α for 2 h. (**A**) The expression of NF-κB p65 subunit mRNA detected by qPCR. (**B**) NF-κB nuclear/cytosolic protein ratio examined by western blot. (**C**) mRNA expression levels of inflammatory cytokine IL-1β. (**D**) mRNA expression levels of adhesion molecule ICAM-1. *n* = 6 in each group. + *p* ≤ 0.05 compared to control. * *p* ≤ 0.05 compared to TNF-α-treated cells.

**Figure 2 ijms-23-03065-f002:**
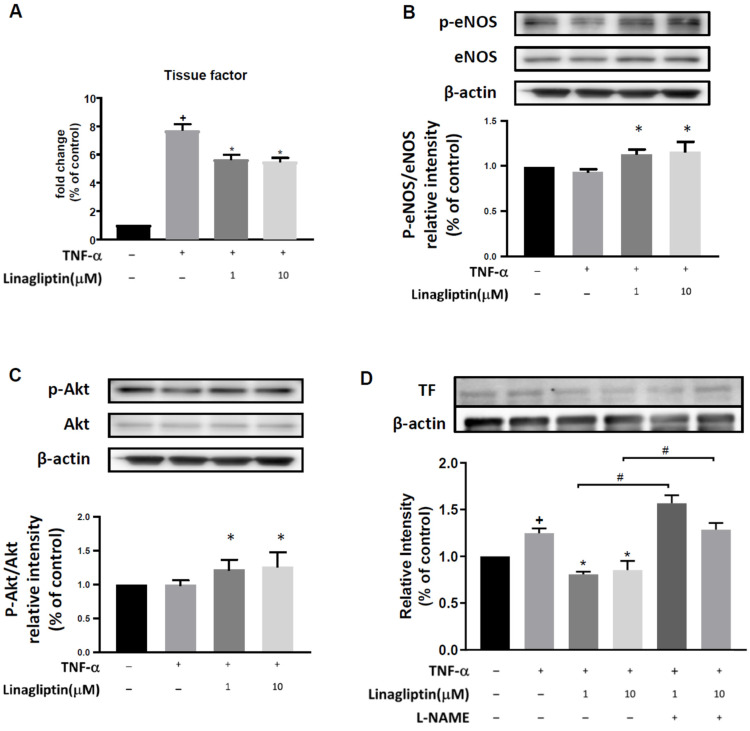
Linagliptin suppresses TNF-α-induced tissue factor expression via an Akt/eNOS pathway. HUVECs were pretreated with different doses (1 and 10 µM) of linagliptin for 48h and then incubated with 0.5 ng TNF-α for 2h. Gene expression and protein levels were detected by qPCR and western blot. (**A**) Expression levels of tissue factor (TF). (**B**) Phosphorylation ratio of eNOS. (**C**) Phosphorylation ratio of Akt. (**D**) Protein levels of TF with or without L-NAME pretreatment, an eNOS inhibitor. *n* = 6 in each group. + *p* ≤ 0.05 compared to respective control. * *p* ≤ 0.05 compared to TNF-α-treated cells. # *p* ≤ 0.05 compared to TNF-a plus linagliptin-treated group.

**Figure 3 ijms-23-03065-f003:**
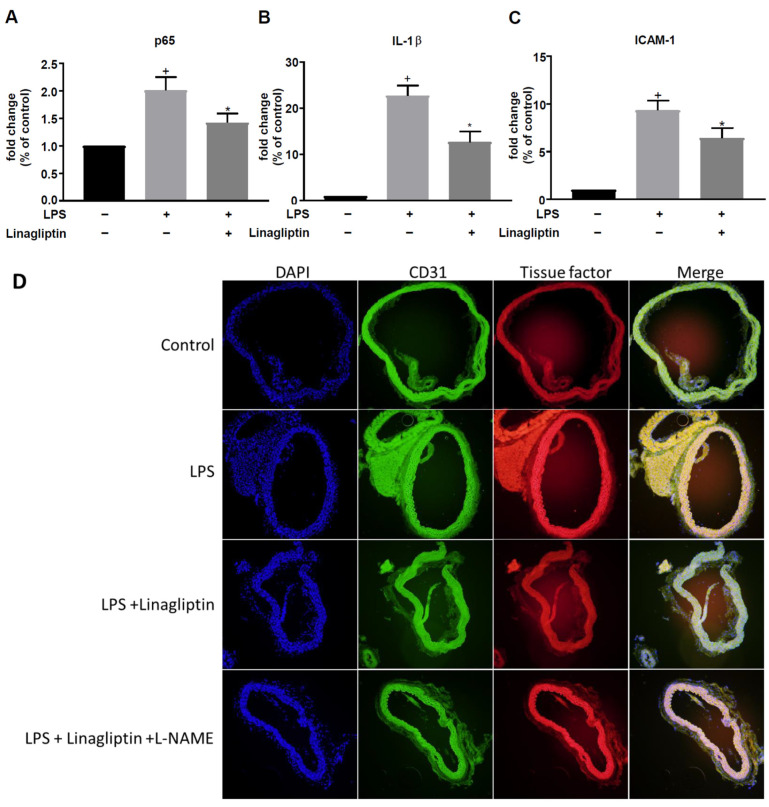
Linagliptin attenuated the LPS-induced expression of inflammatory cytokines in vivo. The aortic tissues of LPS-injected mice with or without linagliptin treatment were isolated to measure the fold change in gene expression of (**A**) p65, (**B**) IL-1β, and (**C**) ICAM-1 by qPCR. (**D**) Immunofluorescence stain of CD-31 and TF in cross-sections of mouse aorta. *n* = 7 in each group. + *p* ≤ 0.05 compared to untreated controls. * *p* ≤ 0.05 compared to LPS-treated mice.

**Figure 4 ijms-23-03065-f004:**
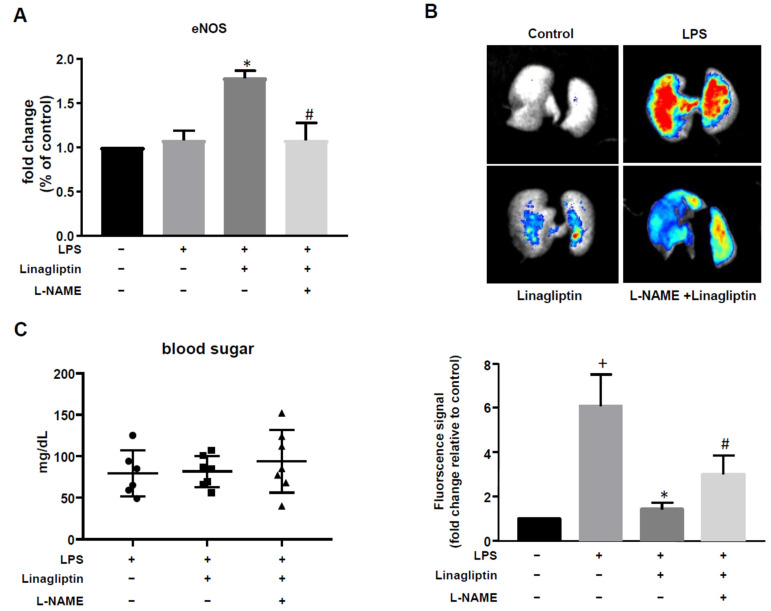
Linagliptin increased eNOS expression and suppressed microvascular thrombosis. L-NAME was added into feeding water for 2 weeks before linagliptin treatment. Mice were fed linagliptin 2 days before LPS injection. (**A**) The fold change in eNOS expression was measured by qPCR. (**B**) Microvascular thrombosis was detected by fluorescence imaging using fluorescent microbeads. (**C**) Blood sugar levels of LPS-injected mice with or without L-NAME or linagliptin treatment. *n* = 7 in each group. + *p* ≤ 0.05 compared to control. * *p* ≤ 0.05 compared to LPS-treated group. # *p* ≤ 0.05 compared to LPS plus linagliptin-treated group.

**Figure 5 ijms-23-03065-f005:**
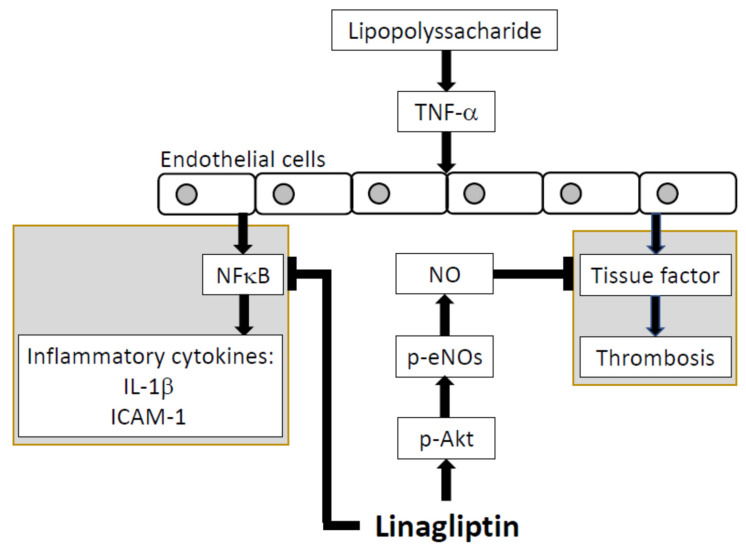
Proposed mechanism. Through inhibition of the NFκB p65 subunit, linagliptin can down-regulate IL-1β and ICAM-1 expression. By increasing eNOS expression, linagliptin can ameliorate tissue factor activity.

## Data Availability

The data presented in this study are available on request from the corresponding author.

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
