# Peer review of "The Dipeptidyl Peptidase-4 Inhibitor Linagliptin Ameliorates Endothelial Inflammation and Microvascular Thrombosis in a Sepsis Mouse Model"

_ijms, 2022, doi:10.3390/ijms23063065_

Round 1

Reviewer 1 Report

Authors have properly responded to my comments and suggestions.

Author Response

Thanks for your advice.

Our manuscript has received English editing for resubmission.

Reviewer 2 Report

Wang et al reproduced data made with linagliptin about 10 (Kroller-Schon) and 5 (Steven) years ago, but add value with a mechanistic (in vivo) explanation supporting the NO hypothesis, responsible for  positive effect of linagliptin in  exp. sepsis.

Major points: As the authors pointed out by themselves, a major limitation of the in vitro part of the  study are the used concentration of a 1nM inhibitor in 1µM or 10 µM concentrations! As this very high and also results do not show any concentration responses, authors have to discuss potential selectivity issues of  this drug in the 1µM and beyond range. 

In vivo: the study is performed with an appropriate dose of 5 mg/kg/d. However no basic in vivo parameters are reported such as body weight and survival; please include those data and test for significance, too. This is relevant regarding the severity grade of the model. In addition, no data on DPP-4 inhibition (e.g. from plasma) or GLP-1 is shown, please add one of those data sets to the result part. Are any other in vivo data on plasma cytokines available? I recommend to present the in vivo data mainly in scatter blot rather than simple box blots to allow to demonstrate variations of the individual data. 

Minor: Authors speculate on clinical use of this drug for sepsis; please put in context the dose you  used in your study to the clinical dose used; e.g. cite plasma cmax and discuss potential posology.

Methods: Please state in the material section where linagliptin was purchased, from; what was the vehicle solution, which concentration? 

Author Response

Major points: As the authors pointed out by themselves, a major limitation of the in vitro part of the study are the used concentration of a 1nM inhibitor in 1µM or 10 µM concentrations! As this very high and also results do not show any concentration responses, authors have to discuss potential selectivity issues of this drug in the 1µM and beyond range.

Ans: Considering the growth condtion of HUVECs in plates, we could only observe the drug effects for a limited time. Therefore, we used high dose linagliptin (1µM or 10 µM) and L-NAME (1nM). We apologized for that our study was not designed for finding appropriate dose of linagliptin in treating sepsis. Further studies are needed to explore dose response relationship in TNF-a treated HUVECs.

In vivo: the study is performed with an appropriate dose of 5 mg/kg/d. However no basic in vivo parameters are reported such as body weight and survival; please include those data and test for significance, too. This is relevant regarding the severity grade of the model.

Ans: Thanks for your suggestion. In LPS and L-NAME pretreated groups, the mice body weight changed significantly but our mice all survived 24hrs after LPS injection. We have presented our body weight changes data in supplementary data.

In addition, no data on DPP-4 inhibition (e.g. from plasma) or GLP-1 is shown, please add one of those data sets to the result part. Are any other in vivo data on plasma cytokines available?

Ans: To harvest mice lungs for pulmonary fluorescence imaging, we had to inflate the lung via 24G catheter immediately after sacrifice. In order not to cause thrombus formation, we tended to take blood samples after harvesting the lung. Unfortunately, the volume of blood samples was low and we could only use these samples for blood glucose test. We apologized for that we could only show the inflammatory cytokines levels in aortic tissue.

I recommend to present the in vivo data mainly in scatter blot rather than simple box blots to allow to demonstrate variations of the individual data.

Ans: thanks for your advice, we have revised our figure 4C with scatter plot.

Minor: Authors speculate on clinical use of this drug for sepsis; please put in context the dose you used in your study to the clinical dose used; e.g. cite plasma cmax and discuss potential posology.

Ans: Our study intended to explore the possible mechanism for Linagliprin protective effects in sepsis, which might impact clinicians in drug selection for glucose control in diabetic patients. Further clinical investigations are needed to establish the survival benefits and appropriate dose of Linagliptin in treating sepsis.

Methods: Please state in the material section where linagliptin was purchased, from; what was the vehicle solution, which concentration?

Ans: Thanks for your comments, we have added this information in our materials and method section line 88, 90, 98 and 99.

Round 2

Reviewer 2 Report

thanks for clarification of the open questions; (for DPP-4 activity testing only 1 µl of plasma is needed....)

This manuscript is a resubmission of an earlier submission. The following is a list of the peer review reports and author responses from that submission.

Round 1

Reviewer 1 Report

This manuscript authored by Wang SC, et al. indicates that linagliptin, a DPP-4 inhibitor, reduces endothelial inflammation and microvascular thrombosis in a sepsis mouse model. It may have translational implication.   

Major concerns:

  1. Hyperproduction of nitric oxide (NO) by upregulation of iNOS contributes to hypotension and organ failure. LPS and TNF-alpha are potent molecules to stimulate production of NO. Constitutive endothelial NOS (eNOS) is involved in formation of NO in septic shock as well. Knockout mice without eNOS were resistant to septic shock [Connelly L, et al. J Biol Chem 2005; 280: 10040-10046]. LPS increased formation of eNOS and p-AKT in HUVECs time-dependently. The results in the submitted manuscript reported that p-eNOS and pAkt were unchanged after treatment of TNF-alpha in HUVECs. Expression of eNOS remained unchanged in LPS-induced sepsis mice as well. The discrepancy about eNOS in sepsis model needs to be explained.   
  2.  The rationale to use only one dose of linagliptin (5 mg/kg/day) need to be provided. 

Author Response

Reviewer 1

This manuscript authored by Wang SC, et al. indicates that linagliptin, a DPP-4 inhibitor, reduces endothelial inflammation and microvascular thrombosis in a sepsis mouse model. It may have translational implication.

Q1. Hyperproduction of nitric oxide (NO) by upregulation of iNOS contributes to hypotension and organ failure. LPS and TNF-alpha are potent molecules to stimulate production of NO. Constitutive endothelial NOS (eNOS) is involved in formation of NO in septic shock as well. Knockout mice without eNOS were resistant to septic shock [Connelly L, et al. J Biol Chem 2005; 280: 10040-10046]. LPS increased formation of eNOS and p-AKT in HUVECs time-dependently. The results in the submitted manuscript reported that p-eNOS and pAkt were unchanged after treatment of TNF-alpha in HUVECs. Expression of eNOS remained unchanged in LPS-induced sepsis mice as well. The discrepancy about eNOS in sepsis model needs to be explained.

Ans: Thanks for the comments. We agreed with you that eNOS plays a complicated role in sepsis. Decreased eNOS activity in septic mice contributed to a poorer outcome1. In an observational study, sepsis perturbated endothelial function and NO production2. Microvascular dysfunction due to reduced eNOS activity and NO synthesis also appears to play an important role in outcomes3. Our study was not designed to elucidate such complicated network regulating eNOS in sepsis. As you suggested, we added this point into our discussion. (Please see the page 20 line 7-12)

Q2. The rationale to use only one dose of linagliptin (5 mg/kg/day) need to be provided.

Ans: Thanks for the comments. As you suggested, we provided the rationale to use of the dose of linagliptin (5 mg/kg/day) in the current study. Please see the reference #4 in the manuscript.

Thank you very much again. We appreciate your comments!

Reviewer 2 Report

The manuscript titled The dipeptidyl peptidase-4 inhibitor linagliptin ameliorates endothelial inflammation and microvascular thrombosis in a sepsis mouse model by Wang et al. describes the protection provided by linagliptin against inflammatory and thrombotic effects caused by either TNFa in human umbilical vein endothelial cells or LPS-injected mice. They propose the inhibition of NFKb dependent IL-1b and ICAM-1, and the inhibition of TF and pulmonary thrombosis by the Akt/eNOS pathway. However, this manuscript is very much deficient in terms of the quality and the technology used to obtain the results presented that are insufficient to support the proposed conclusions.

Major

  1. Figures 1 and 2 refer to in vitro experiments with HUVEC showing nice statistics. However, the number of independent experiments used to apply statistics are missing. Figures 3 and 4 refer to in vivo experiments but no indications about the number of animals used in each graph.
  2. Figure 1B shows WB of nuclear and cytosolic levels of p65, a component of NFKb. However, there is not any description of the protocols used to isolated nuclei. The whole original blots should be included as complementary figures. This is a very limited demonstration of NFKb activation of IL-1b and ICAM-1. DNA binding assays (ChIP, EMSA) should be included. Protein levels of IL-1b and ICAM-1 should be also included.
  3. Tissue factor activation by Akt/eNOS pathway should be better demonstrated. For example, specific silencing of eNOS and Akt should be induced in HUVEC to demonstrate their implications. Also, which is the effect of L-NAME by itself on to TF RNA and protein levels?
  4. I am concern about the term “aortic tissues” used for experiments in figure 3. This analysis should be done in aortic endothelium to fit with the previous results shown in HUVEC. Also, protein levels should be also included.
  5. Figure 4 A should be completed with the RNA binding assay. Fig. 4B: Which is the effect of L-NAME itself in microvascular thrombosis? Fig. 4C is very poor. A complete description of glucose determination conditions should be explained, e.g. fasting period, animal control without treatment, the effect of glucose levels under the different isolated treatments,…

Minor

The edition of this manuscript is very much deficient, and it looks like the authors have submitted a previous draft without revision. See lines 236 and 237. See also Author contributions. Authors indicate in M&M the existence of the approval of ethical committee. This should be also included in Institutional Review Board Statement.

A deep english editing should be also carried out.

Author Response

Reviewer 2

The manuscript titled: The dipeptidyl peptidase-4 inhibitor linagliptin ameliorates endothelial inflammation and microvascular thrombosis in a sepsis mouse model by Wang et al. describes the protection provided by linagliptin against inflammatory and thrombotic effects caused by either TNFa in human umbilical vein endothelial cells or LPS-injected mice. They propose the inhibition of NFKb dependent IL-1b and ICAM-1, and the inhibition of TF and pulmonary thrombosis by the Akt/eNOS pathway. However, this manuscript is very much deficient in terms of the quality and the technology used to obtain the results presented that are insufficient to support the proposed conclusions.

Ans: Thanks for the comments. We agree with you that there are some deficits in the current study. However, using this simple animal disease model, we provided first evidence that the DPP-4 inhibitors (linagliptin) could exert protective effects against endothelial inflammation and microvascular thrombosis in sepsis mice. These findings provide crucial pre-clinical data and could be tested in further large clinical study.

Major

Q1. Figures 1 and 2 refer to in vitro experiments with HUVEC showing nice statistics. However, the number of independent experiments used to apply statistics are missing. Figures 3 and 4 refer to in vivo experiments but no indications about the number of animals used in each graph.

Ans: Thanks for the comments. As you suggested, we provided the number of independent experiments used to apply statistics in the study. For cell experiments, there were six samples in each group, and, there were seven in each group for animal experiments. We have added these details in the figure legends

Q2. Figure 1B shows WB of nuclear and cytosolic levels of p65, a component of NFKb. However, there is not any description of the protocols used to isolated nuclei. The whole original blots should be included as complementary figures. This is a very limited demonstration of NFKb activation of IL-1b and ICAM-1. DNA binding assays (ChIP, EMSA) should be included. Protein levels of IL-1b and ICAM-1 should be also included.

Ans: Thanks for the comments. For experiments involving nuclear proteins, cytosolic and nuclear proteins were separated using a nuclear/cytosol fractionation kit (BioVision, Milpitas, CA, USA) and their concentrations were measured using a Bradford protein assay (Bio-Rad, Hercules, CA, USA). As you suggested, we have added such description in the method section. (Please see the page 9 line 7-10)

Q3. Tissue factor activation by Akt/eNOS pathway should be better demonstrated. For example, specific silencing of eNOS and Akt should be induced in HUVEC to demonstrate their implications. Also, which is the effect of L-NAME by itself on to TF RNA and protein levels?

Ans: Thanks for the comments. We agreed with you that tissue factor activation by Akt/eNOS pathway should be better demonstrated. Unfortunately, we still try p-Akt inhibitor western blot and are still working on this. As you suggested, we mentioned this point in the study limitation. (Please see the page 20 line7-10)

Q4. I am concern about the term “aortic tissues” used for experiments in figure 3. This analysis should be done in aortic endothelium to fit with the previous results shown in HUVEC. Also, protein levels should be also included.

Ans: Thanks for the comments. However, mice endothelium is too few for the analysis by western blotting. In order to minimize experimental animal sacrifice, we used immunofluorescence staining of CD-31 and tissue factor in the study.

Q5. Figure 4 A should be completed with the RNA binding assay. Fig. 4B: Which is the effect of L-NAME itself in microvascular thrombosis? Fig. 4C is very poor. A complete description of glucose determination conditions should be explained, e.g. fasting period, animal control without treatment, the effect of glucose levels under the different isolated treatments.

Ans: Thanks for the comments. As you suggested, we provided the data of the effect of L-NAME in microvascular thrombosis and also improved the Fig. 4C. We agreed with you and provided the information about a complete description of glucose determination conditions in the current study. (Please see the page 11 line 14-19)

Minor

Q6. The edition of this manuscript is very much deficient, and it looks like the authors have submitted a previous draft without revision. See lines 236 and 237. See also Author contributions. Authors indicate in M&M the existence of the approval of ethical committee. This should be also included in Institutional Review Board Statement. A deep English editing should be also carried out.

Ans: Thanks for the comments. As you suggested, we revised some deficits in the manuscript.

Thank you very much again. We appreciate your comments!

Reviewer 3 Report

The authors present a human endothelial cell model (human umbilical vein endothelial cells, HUVECs) of sepsis using LPS induced inflammation and compare the effects of the competitive, reversible  Dipep-23 tidyl peptidase (DPP)-4 inhibitor Linagliptin on inflammation and coagulation by looking at such factors such as TNF-α induced NFkB, Il-1b, ICAM_1 and Tissue Factor.

The authors are able to show that linagliptin treatment with of HUVEC has anti-inflammatory and antithrombotic effects. The conclude that treatment with linagliptin could be a potential adjuvant treatment to protect against sepsis induced disseminated intravascular coagulation (DIC).

As sepsis, despite improvements in outcomes with bundled care, remains a disease with high mortality and morbidity, new treatment approaches are needed to continue improve care. Unfortunately, the current landscape of sepsis care appears to have a disconnect between what is seen in the laboratory and in clinical reality. To refocus the research community and provide clinical meaningful research a working group of the Society of Critical Care Medicine and the European Society of Intensive Care Medicine have published several articles regarding the “Research Priorities for Sepsis Research”. (1, 2)

The submitted article attempts to address some of these items however some aspects appear oversimplified.

With the new sepsis definition, the focus has moved away from simple understanding that sepsis causes hyperinflammation but is rather a dysregulation were the checks and balances of pro and anti-inflammation and/ or coagulation are in disarray. (3, 4) The authors should highlight this concept a little more in their introduction and their limitation discussion. Articles by Osuchowski and Remick DG et al focus on this concept and the importance of this mixed inflammatory and coagulation response. (5, 6)

OTHER

The following should be considered

Page 2 line 50 and 51: Source control with the administration of antimicrobial or antiviral medication along with surgical source control is an important pilar of clinical sepsis care and is part of the sepsis care bundles and should be added. (7, 8)

Page 2 Line 56-57: similar to the above concern regarding the simplicity of the description of sepsis as pro-inflammatory and pro-coagulation the authors should acknowledge that this is rather a dysregulated response of the immune system and coagulation system for which pro and anti-responses are disorganized. Understanding one arm of this disorganized system such as the pro-inflammation is important but need to be put in context.

Page 3 Line 100: please spell once L-Name as NG-nitro-L-arginine-methylester.

Page 10 to 11, Line 328 to 373 à the submitted version is missing this information and just contains journal template. Please complete.

  1. Deutschman CS, Hellman J, Ferrer Roca R, et al. The Surviving Sepsis Campaign: Basic/Translational Science Research Priorities. Crit Care Med 2020;48(8):1217-1232.
  2. Deutschman CS, Hellman J, Roca RF, et al. The surviving sepsis campaign: basic/translational science research priorities. Intensive Care Med Exp 2020;8(1):31.
  3. Seymour CW, Liu VX, Iwashyna TJ, et al. Assessment of Clinical Criteria for Sepsis: For the Third International Consensus Definitions for Sepsis and Septic Shock (Sepsis-3). JAMA 2016;315(8):762-774.
  4. Singer M, Deutschman CS, Seymour CW, et al. The Third International Consensus Definitions for Sepsis and Septic Shock (Sepsis-3). JAMA 2016;315(8):801-810.
  5. Osuchowski MF, Craciun F, Weixelbaumer KM, et al. Sepsis chronically in MARS: systemic cytokine responses are always mixed regardless of the outcome, magnitude, or phase of sepsis. J Immunol 2012;189(9):4648-4656.
  6. Osuchowski MF, Welch K, Siddiqui J, et al. Circulating cytokine/inhibitor profiles reshape the understanding of the SIRS/CARS continuum in sepsis and predict mortality. J Immunol 2006;177(3):1967-1974.
  7. Vincent JL. Clinical sepsis and septic shock--definition, diagnosis and management principles. Langenbecks Arch Surg 2008;393(6):817-824.
  8. Marshall JC, al Naqbi A. Principles of source control in the management of sepsis. Crit Care Clin 2009;25(4):753-768, viii-ix.

Author Response

Reviewer 3

The authors present a human endothelial cell model (human umbilical vein endothelial cells, HUVECs) of sepsis using LPS induced inflammation and compare the effects of the competitive, reversible Dipeptidyl peptidase-4 (DPP)-4 inhibitor Linagliptin on inflammation and coagulation by looking at such factors such as TNF-α induced NFkB, Il-1b, ICAM_1 and Tissue Factor.

The authors are able to show that linagliptin treatment with of HUVEC has anti-inflammatory and antithrombotic effects. The conclude that treatment with linagliptin could be a potential adjuvant treatment to protect against sepsis induced disseminated intravascular coagulation (DIC).

As sepsis, despite improvements in outcomes with bundled care, remains a disease with high mortality and morbidity, new treatment approaches are needed to continue improve care. Unfortunately, the current landscape of sepsis care appears to have a disconnect between what is seen in the laboratory and in clinical reality. To refocus the research community and provide clinical meaningful research a working group of the Society of Critical Care Medicine and the European Society of Intensive Care Medicine have published several articles regarding the “Research Priorities for Sepsis Research”.

The submitted article attempts to address some of these items however some aspects appear oversimplified.

Q1. With the new sepsis definition, the focus has moved away from simple understanding that sepsis causes hyperinflammation but is rather a dysregulation were the checks and balances of pro and anti-inflammation and/ or coagulation are in disarray. The authors should highlight this concept a little more in their introduction and their limitation discussion. Articles by Osuchowski and Remick DG et al focus on this concept and the importance of this mixed inflammatory and coagulation response.

Ans: Thanks for the comments. As you suggested, we had added these in our first paragraph of the introduction and references.

OTHER

The following should be considered

Q2. Page 2 line 50 and 51: Source control with the administration of antimicrobial or antiviral medication along with surgical source control is an important pilar of clinical sepsis care and is part of the sepsis care bundles and should be added. (7, 8)

Ans: Thanks for the comments. We agreed with you that the administration of antimicrobial or antiviral medication along with surgical source control is an important pilar of clinical sepsis care and should be added. Unfortunately, our study design cannot provide such information. As you suggested, we mentioned this point in our first paragraph of introduction.

Q3. Page 2 Line 56-57: similar to the above concern regarding the simplicity of the description of sepsis as pro-inflammatory and pro-coagulation the authors should acknowledge that this is rather a dysregulated response of the immune system and coagulation system for which pro and anti-responses are disorganized. Understanding one arm of this disorganized system such as the pro-inflammation is important but need to be put in context.

Ans: Thanks for the comments. As you suggested, we acknowledge that the sepsis model is rather a dysregulated response of the immune system and coagulation system for which pro and anti-responses are disorganized. We mentioned this point in our first paragraph of introduction.

Q4. Page 3 Line 100: please spell once L-Name as NG-nitro-L-arginine-methylester.

Ans: Thanks for the comments. As you suggested, we revised the L-Name as NG-nitro-L-arginine-methylester at the 3rd line on page.

Q5. Page 10 to 11, Line 328 to 373 à the submitted version is missing this information and just contains journal template. Please complete.

Ans: Thanks for the comments. As you suggested, we complete the missing information

Thank you very much again. We appreciate your comments!

Round 2

Reviewer 1 Report

This is a revised manuscript submitted by Wang SC, et al. Authors have addressed previous questions and did changes in the revised manuscript. Some new references were added.

Major concerns with the revised manuscript:

  1. Research Design: In vitro study was carried out in human umbilical vein endothelial cells (HUVEC). In vivo study was performed in a mouse model. Please explain the rationale. 
  2. In Method Section, Line 94, 1 and 10 µM of linagliptin had no effect on cell viability. In Line 96, 1 and 10 ng of linagliptin was used. Authors need to clarify the doses of lingliptin.
  3. In Result Section, Tissue factor expression was detected. Gene transcription of tissue factor was about 8-fold increase in TNF-alpha group compared to control in Fig. 2A. However, protein expression of tissue factor was only around 1.2 -fold increase in TNF-alpha group in Figure 2D. Please explain this discrepancy.  

Author Response

Responses to reviewer’s comments

Thank you very much for your review of our manuscript, which contributed to improving its quality. These comments are very instructive and helpful to our research. We had tried our best to reply to your comments point-by-point and revised the manuscript accordingly. The responses to your comments are dictated below. Modifications made in the new version of the manuscript are highlighted in red.

Reviewer 1:

Major concerns with the revised manuscript:

Q1. Research Design: In vitro study was carried out in human umbilical vein endothelial cells (HUVEC). In vivo study was performed in a mouse model. Please explain the rationale. 

Ans: Thanks for the comments. We agree with your concern. Mouse models have been used extensively to provide insight into the mechanisms underlying many diseases, explore the efficacy of candidate drugs and predict patient responses (1). Therefore, we use the mouse model to verify the in vitro findings in HUVECs.

Q2. In Method Section, Line 94, 1 and 10 µM of linagliptin had no effect on cell viability. In Line 96, 1 and 10 ng of linagliptin was used. Authors need to clarify the doses of linagliptin.

Ans: Thanks for the comments. We apologize for our typographical error. As you suggested, we have fixed this in line 96.

Q3. In Result Section, Tissue factor expression was detected. Gene transcription of tissue factor was about 8-fold increase in TNF-alpha group compared to control in Fig. 2A. However, protein expression of tissue factor was only around 1.2 -fold increase in TNF-alpha group in Figure 2D. Please explain this discrepancy.  

Ans: Thanks for the comments. The limited correlations observed between mRNA and protein abundance suggest pervasive regulation of post-transcriptional steps and support the importance of profiling mRNA levels in parallel to protein synthesis and degradation rates(2). We agree with you that our study was not designed to focus on regulatory mechanism in tissue factor’s mRNA translation.

References

  1. Justice MJ, and Dhillon P. Using the mouse to model human disease: increasing validity and reproducibility. Dis Model Mech. 2016;9(2):101-3.
  2. Aviner R, Shenoy A, Elroy-Stein O, and Geiger T. Uncovering Hidden Layers of Cell Cycle Regulation through Integrative Multi-omic Analysis. PLoS Genet. 2015;11(10):e1005554.

Thank you very much again. We appreciate your comments!